# A Network Analysis of the Relationship among Reading, Spelling and Maths Skills

**DOI:** 10.3390/brainsci11050656

**Published:** 2021-05-18

**Authors:** Pierluigi Zoccolotti, Paola Angelelli, Chiara Valeria Marinelli, Daniele Luigi Romano

**Affiliations:** 1Department of Psychology, University of Roma “Sapienza”, 00185 Rome, Italy; 2Neuropsychology Unit, IRCCS Fondazione Santa Lucia, 00179 Rome, Italy; 3Department of History, Society and Human Studies, Lab of Applied Psychology and Intervention, University of Salento, 73100 Lecce, Italy; paola.angelelli@unisalento.it (P.A.); chiaravaleria.marinelli@unisalento.it (C.V.M.); daniele.romano@unimib.it (D.L.R.); 4Department of Clinical and Experimental Medicine, University of Foggia, 71122 Foggia, Italy; 5Department of Psychology, NeuroMi Milan Centre of Neuroscience, University of Milano-Bicocca, 20126 Milan, Italy

**Keywords:** reading, spelling, maths

## Abstract

**Background.** Skill learning (e.g., reading, spelling and maths) has been predominantly treated separately in the neuropsychological literature. However, skills (as well as their corresponding deficits), tend to partially overlap. We recently proposed a multi-level model of learning skills (based on the distinction among competence, performance, and acquisition) as a framework to provide a unitary account of these learning skills. In the present study, we examined the performance of an unselected group of third- to fifth-grade children on standard reading, spelling, and maths tasks, and tested the relationships among these skills with a network analysis, i.e., a method particularly suited to analysing relations among different domains. **Methods.** We administered a battery of reading, spelling, and maths tests to 185 third-, fourth-, and fifth-grade children (103 M, 82 F). **Results.** The network analysis indicated that the different measures of the same ability (i.e., reading, spelling, and maths) formed separate clusters, in keeping with the idea that they are based on different competences. However, these clusters were also related to each other, so that three nodes were more central in connecting them. In keeping with the multi-level model of learning skills, two of these tests (arithmetic facts subtest and spelling words with ambiguous transcription) relied heavily on the ability to recall specific instances, a factor hypothesised to underlie the co-variation among learning skills. **Conclusions.** The network analysis indicated both elements of association and of partial independence among learning skills. Interestingly, the study was based on standard clinical instruments, indicating that the multi-level model of learning skills might provide a framework for the clinical analysis of these learning skills.

## 1. Introduction 

The challenge of modelling reading, spelling, and maths skills (and their corresponding impairments) has received great attention. For example, in the case of reading, models, such as the dual-route cascaded model (DRC [1]), the triangle model [2,3], and the CDP+ model [4], provide different alternatives in terms of architecture (with either one or two mechanisms) and operating features, but share the focus on reading behaviour alone. Similar approaches have been used in the cases of spelling [5,6] and maths [7].

In general, these models focus on the independence of these skills (and of their impairments). Indeed, in the last 40 years, there has been increasing emphasis on the specificity of learning disorders, i.e., dyslexia, dysgraphia, and dyscalculia, which has also influenced international manuals, such as the DSM-IV or the ICD-10. However, more recent literature has also recognised that learning disorders tend to be associated. Thus, children with dyslexia tend to have a much higher probability of also having spelling and maths difficulties compared to the general population [8,9]. Therefore, if we attempt to model them separately, we risk losing important parts of the phenomenon. By contrast, studying learning disorders from the perspective of comorbidity may help place reading, spelling, and maths skills within a unitary framework. 

In a recent study [10,11], we examined reading, spelling, and maths skills in an unselected population of children and examined a set of predictors of these learning behaviours. Notably, we observed significant overlap among predictors; predictors of reading also predicted spelling and maths skills quite well (although not perfectly). Conversely, predictors of spelling also predicted reading and maths skills; and, finally, predictors of maths effectively predicted reading and spelling. When the effect of specific predictors was examined in greater detail, we found that some were specifically associated with only one behaviour, others with different behaviours but only for some parameters (such as speed but not accuracy), and some were associated with all learning skills. 

To account for these findings, we propose that a comprehensive model of the cognitive processes underlying reading, spelling and maths should be framed at different levels of analysis. Starting from Chomsky’s [12] original proposal, we propose articulating the factors relevant for learning skills in terms of “competence” and “performance”. However, for reasons to be presented below, we also included a separate “acquisition” level. The major characteristics of the competence, performance and acquisition levels are summarised in Table 1. In keeping with Chomsky’s [12] original proposal, competence refers to the ability to activate a specific set of representations and processes. Competence captures the core aspects of a given behaviour which is expressed across different tasks and conditions. Major models of reading can be seen as an attempt to describe the core reading competence (for a discussion of this point, see Bishop [13]). Accordingly, different competencies may be required for reading, spelling, and maths. A multi-level model of learning skills that assumes these separate competences is presented in Figure 1. In this perspective, deficits in core competences (or developmental failures in acquiring them) would account for the dissociations of symptoms reported in the acquired and, to a lesser extent, in the developmental literature.

In Chomsky’s [12] terms, performance refers to the fact that skills (or deficits) in a given domain do not directly refer only to a given competence, but also reflect the additional influence of task factors. Indeed, behaviours can only be examined through specific tasks. Thus, according to Chomsky’s [12] proposal, in naturalistic conditions every measure of a given skill (such as reading, spelling, or maths) reflects the combined effect of competence and sensitivity to task factors (performance). In general, performance factors are omitted by major traditional cognitive models of reading (as well as spelling and maths), such as those quoted above [1,2,3,4,5,6,7]. However, there is now a parallel tradition in developing models that explicitly address the eye movement control in reading, such as the EZ reader model [14] or the OB1-reader [15]. These reading models can be seen as an attempt to merge competence factors with the requirement to account for actual performance in reading, particularly taking into account the specific role of eye movements. In the model in Figure 1, performance factors may contribute to both specificity and association of performance. For example, the ability to integrate task sub-components (as measured by the well-known RAN tasks) may not only influence the ability to read but also that of efficiently carrying out complex mental calculations. Notably, it has been reported that RAN tasks predict fluency in reading and making calculations but not accuracy in either of these two domains [16]. Thus, performance factors may be closely tied to specific parameters, such as speed or accuracy. 

We have proposed that a level of acquisition should be added to Chomsky’s [12] original dichotomous scheme. Acquisition expresses itself as the effect of practice influencing behaviours in various ways. Firstly, it is necessary to let a given competence emerge (even if this is partially pre-wired as in the case of language). Secondly, practice is also necessary to tune a given skill in specific task formats, such as reading from left to right. Thirdly, practice is important for learning specific memory traces (or instances). This capacity is critical for automatising behaviours passing from algorithm-based responding to the direct, as well as obligatory, retrieval of specific memory traces [17,18]. For example, in real life, computations are typically not carried out using arithmetic algorithms, but through strategic processing, largely relying on the knowledge of arithmetic facts, such as tables. It has been proposed that the capacity to learn and activate specific memory traces acts as a domain-independent factor [11]. Neurophysiological evidence indicates that the ability to use specific memory traces is a general skill that favours behavioural efficiency closely associated with hippocampal functioning [19]. Accordingly, automatization might contribute to the overlap between reading, spelling, and maths skills and, conversely, to explaining the comorbidity among dyslexia, dysgraphia, and dyscalculia. We have proposed that automatization through the activation of specific memory traces goes beyond the competence–performance dichotomy proposed by Chomsky [12], thus requiring an independent level of analysis [11].

The model illustrated in Figure 1 may help to frame the relationships among reading, spelling, and maths skills (as well as their relative impairments). As briefly outlined above, there are reasons for predicting some degree of dissociation between these skills, as well as some degree of association. The important point here is that the model could be instrumental for understanding the sources of overlap and of partial independence.

In the present study, we set out to study the association between various tests of reading, spelling, and maths, using, as a reference, the multi-level model of learning skills presented in Figure 1. We focused on tests that are used widely in clinical settings which, as to be expected, are loaded to variable extents with the components envisaged by the model (i.e., competence, performance, and acquisition). Thus, one would expect that tests which map on the same competence (e.g., reading) would tend to be intercorrelated but poorly correlated with tests which map on different competences (e.g., spelling or maths). Additionally, one would expect that, over and above the influence of competence factors, tests based on similar formats or performance measures (e.g., speed) would tend to be intercorrelated, whereas tests based on dissimilar formats and/or measures would tend to be uncorrelated. Finally, the model predicts that tasks which heavily require the ability to recall specific instances would be correlated, regardless of their specific domain. For example, the acquisition of arithmetic facts (such as tables) requires the ability to consolidate specific instances which, in the case of maths, enable speeding up performance in mental or written calculations by avoiding the time-consuming algorithm processing. In the case of spelling, it is well known that some Italian words (such as “quota” (rate) or “cuoco” (chef)) have an ambiguous transcription [20]. In this case, one would expect the child to acquire the correct orthographic form through repeated exposition to printed materials [21]. The model predicts that performance on tasks calling for instance-based retrieval, such as recalling arithmetic facts and spelling words with ambiguous transcription, should be positively inter-correlated over and above the effect of the competence and performance factors.

In pursuing these aims, we used a method that is particularly suited to analysing relationships among different domains, i.e., the network analysis. This is a data-driven approach that is especially useful for exploring domains where many variables are correlated, and their relationships are the core information to be examined. A network is a model composed of a set of nodes representing entities and a set of edges that connect the nodes, which represent their relations [22]. To estimate the network, we adopted a Gaussian Graphical Model (GGM) [23]. The use of GGM leads to a sparse network where the edges represent the partial correlations left between two nodes after conditioning on all other nodes. This means that GGM highlights the direct relationship between two variables, also revealing the indirect paths. Thus, the method gives back a comprehensive, simultaneous picture of the direct and indirect relationships that characterise the whole set of variables [24], thus deepening the meaning of the mere correlation between pairs of variables. 

Overall, we examined the performance of an unselected group of third- to fifth-grade children on standard clinical tests of reading, spelling, and maths. We hypothesised that the network emerging from the intercorrelations among these tests could be interpreted based on the aforementioned multi-level model of learning skills [11].

## 2. Materials and Methods

### 2.1. Participants

We recruited 185 children (103 males and 82 females), ranging in age from 7.8 to 11.2 years (mean age: 9.65 ± 0.95) from three primary schools in the south of Italy. In particular, 68 third-grade children (32 F, 28 M, mean age: 8.56 ± 0.32), 41 fourth-grade children (29 F, 28 M, mean age: 9.50 ± 0.32), and 76 fifth-grade children (30 F, 44 M, mean age: 10.56 ± 0.31) participated in the study. All children performed normally (within 2 SD, weighted for age and class) on an intelligence test (i.e., Raven’s Coloured Progressive Matrices [25] mean z score = −0.15, standard deviation = 0.77; range = −1.91 to +1.35). A total of thirteen children were excluded because they performed below 2 SDs in the intelligence test.

### 2.2. Test Materials

#### 2.2.1. Mathematical Skills

The AC-MT battery [26] was used to assess the students’ mathematical skills. The AC-MT battery consists of several subtasks which derive from neuropsychological models of number processing and calculation [27,28,29]. The subtests used in present work are described below: −Computation. This subtest assesses children’s ability to complete written computational operations (addition, subtraction, multiplication, and division);−Number ordering. This task requires understanding the semantics of numbers and thus evaluating number sense. Series of four numbers are presented and the child has to place them in the correct order (from the highest to the lowest and vice versa);−Arithmetical Facts. This task is used to investigate whether children have stored arithmetical facts and are able to automatically retrieve the results of basic and simple operations from memory. Children are asked to recall several arithmetic facts, each within a 5 s time limit. Responses given after the time limit are considered incorrect.

In all subtests, correct responses were considered. 

#### 2.2.2. Text Reading Task

The participants’ reading level was assessed using a standard reading achievement test which is widely adopted to assess Italian children [30]. The MT consists of a series of meaningful texts (i.e., short stories taken from children’s books) of increasing length from grade 1 to grade 5. The children were asked to read a single text depending on their grade and the period of the school year. In our sample, the number of words increased (i.e., from 168 in Grade 3 to 215 words in Grade 5). Each story was printed in black on white cardboard. None of the texts used for this task were used for the text reading comprehension task. Participants were asked to read the text aloud within a 4 min time limit. In the instructions, the children were asked to read as well as they could, i.e., fluently and making as few errors as possible. Reading speed (number of syllables read/s) and accuracy (number of errors, adjusted for the amount of text read) were considered. The accuracy score takes into account the functional meaning of errors. Each word with a substitution, elision, insertion, or inversion of letters was scored as 1, while spontaneous self-corrections, errors that do not change the meaning of the text, changes in stress assignment, repetitions of the same errors, and hesitations were given a score of 0.5. Test–retest indexes for reading speed, as reported by the manuals, ranged between 0.85 and 0.96.

#### 2.2.3. Text Comprehension Task

The tasks consisted of a series of informative texts [30]. For third to fifth graders, the texts ranged from 226 to 306 words, and their length increased with school grade (a different text was used for each grade). The children were asked to read the text in silence at their own pace and then to answer 12 multiple-choice questions, choosing one of four possible alternatives. There was no time limit, and the children could reread the text as many times as they wished. The final score was calculated as the total number of correct responses. As reported in the manuals, alpha coefficients ranged between 0.61 and 0.83.

#### 2.2.4. Spelling Skills

The participants’ spelling abilities were tested with a standard dictation test (test for the Diagnosis of Orthographic Disorder, DDO-2 [31]). The test has three sections; two require the application of transcribing processes based on one-sound-to-one-letter correspondence, and one requires lexical retrieval. In particular:

Section A: regular words with complete one-sound-to-one-letter correspondence (*n* = 70); Section B (which in the test corresponds to Section D): pseudowords with one-sound-to-one-letter correspondence (*n* = 25); Section C: words with unpredictable transcription along the phonological-to-orthographic conversion route and then requiring lexical spelling (e.g., [kw] in [ˈkwɔta], i.e., quota is transcribed as QUOTA and not *CUOTA) (*n* = 55).

Words and pseudowords were given in separate blocks and in a single quasi-randomised order. The examiner read each item aloud in a neutral tone. The children were asked to repeat each item before writing it down (so that the examiner could be sure that they had understood the item). They were permitted to write in either capital or lower-case letters. No feedback was provided on the accuracy of the written response. The number of stimuli spelled correctly in each section was computed. Self-corrections were accepted.

### 2.3. Procedure

Participants were tested individually in the morning in a quiet room during school hours. The tests were administered following a fixed sequence.

### 2.4. Network Analysis

Network analysis is a useful method for exploring complex patterns of relationships. This type of analysis has been used to investigate different phenomena. In psychology, this have proved useful for studying personality and psychopathology [24,32,33,34,35,36], experimental psychology [37], and also for assessing the correlations across neuropsychological test performances [38,39]. The principle behind the networks is simple: take a set of variables of interest (called nodes) and identify their direct and indirect relationships (named edges). How edges are estimated is what differentiates the different types of network analysis. In psychology, edges are typically estimated with the Gaussian Graphical Model (GGM) [40], which estimates regularised partial correlations. 

The regularisation is carried out by using the graphical “least absolute shrinkage and selection operator”, or LASSO, [41] algorithm. The LASSO is tuned by selecting the best operator through the Extended Bayesian information criterion, which, in turn, is regulated by a parameter γ that was set at 0.25, as suggested in the literature [42]. The LASSO reduces at zero edges with little predictive value [23,43]. The idea is to return to a conservative network by limiting false-positive edges, producing replicable and interpretable results [24]. 

Edges indicate conditional dependency between two nodes net of the other nodes. Thus, this algorithm represents a partial correlation. The use of the LASSO guarantees high specificity (i.e., few false positives), but this may lead to low sensitivity (i.e., not all real edges are detected) [44]. Crucially, the potentially missing edges are the least relevant. The advantage is that GGM networks basically do not produce false positives (i.e., to show an edge that does not exist) [45].

In GGMs, the neighbours of each node can be seen as predictors. The strength centrality (i.e., the sum of absolute values of each edge passing through a node) quantifies the predictability of a node according to the number of its neighbours and the strength of its connections [46].

The stability of the results was checked by means of a bootstrapping procedure (1000 resampling). Bootstrap analysis returns the 95% confidence interval (CI) of each edge and the average edge value over the 1000 resampling. The edges that do not include 0 in their CI are more likely to be replicated and thus represent the most consistent results. Edges where the CI reaches 0 have some likelihood of not being found in a different data collection. Edges that cross 0 (with positive and negative values) are unreliable, because in a different sample, it is somewhat likely that they will have an opposite sign. Thus, strong claims can be made for the first category and more cautious conclusions can be drawn for the second.

The analyses were performed using JASP [47]. JASP (Version 0.14.1) [Computer software]) is software that bases the network modules on the *bootnet* [45] and *qgraph* [48] packages of the R statistical software (R Core Team [49]). 

## 3. Results

Figure 2 represents the best network estimated from the data, representing the relationships among the variables examined. The exact values of all edges, as well as the simple correlations among all variables, are reported in Table 2, which also shows the strength centrality index on the diagonal and the descriptive statistics of each variable. Figure 3 reports the bootstrap results. Additionally, we provided a second, simplified figure of the estimated network (Figure 2, lower panel). This figure reports only the most important edges according to their strength and reliability. Specifically, we plotted only those edges stronger than 0.1, which also corresponded, in our case, to the edges that did not include 0 in their confidence interval (see Figure 3). These edges are those that will most likely be replicated in future studies. The simplified figure is of help in highlighting the key connections within the estimated network; however, it is important to keep in mind that any estimated edge is likely to be true but may be less strong or important for general connectivity.

An inspection of Figure 2 suggests a number of observations: −The different measures of the same ability (reading, writing, and arithmetic) form separate clusters;−These clusters are related to each other at different levels (i.e., different nodes), but there are three nodes that are more central in connecting them. Indeed, if we consider the reliable edges that do not include 0 in their CI (see Figure 2 lower panel), the three clusters are connected only by the nodes with the arithmetic facts subtest, spelling words with ambiguous transcription, and accuracy in reading a text passage. The same nodes emerged as those with higher strength centrality, a measure that highlights the importance of the nodes for the network. 

## 4. Discussion

The results offer insight into the relationships between different academic learning skills. In particular, both elements of association and of partial independence are apparent in the network emerging from the data. Indeed, tests tended to cluster within domains. Thus, reading tasks tended to be intercorrelated with each other, and the same was true for tests of spelling and maths skills. We have proposed that the multi-level model of learning skills [11] may provide a framework for interpreting the network emerging from different learning skills. Indeed, the model predicts that separate, independent competences are present in the cases of reading, spelling, and maths tasks, thus contributing to the partial independence of these skills. Note that the relationships that emerged from the network are direct; thus, they show a level of association after removing all the variance that could be explained by other variables of the model. For example, the association of maths tasks was unique over and beyond what was shared with reading and spelling tasks. From this perspective, the clusters of reading, spelling, and maths are consistent with the idea of independent competences for reading, spelling, and maths. 

However, one should also be aware that performance factors might contribute to partial dissociation among learning skills. Thus, the task characteristics of the three spelling conditions examined here tended to be similar to each other, apart from the specific characteristic of the dictated stimulus to be spelled. This is true, although perhaps to a lesser extent in the case of reading where the child has to deal with a text passage in all three conditions tested, even though in one case the task focused on the accuracy and speed of decoding, and in another on the quality of text comprehension. Similarly, the three mathematical tests shared some surface characteristics, such as the fact that the child had to deal with numbers (and not words), although the specific tasks differed appreciably in their specific requirements. 

It should be considered that it is quite difficult to tease out the role of competence and performance factors in standard clinical tests. Although interpretations tend to focus more often on key core competences, the role of performance factors is inherently difficult to control unless specific manipulations are envisaged. For example, in a series of studies, Berent et al. [50,51] tested the widely accepted hypothesis that dyslexia is associated with a deficit in phonological competence by testing the sensitivity to general phonological contrasts. The results indicated that children with dyslexia showed spare phonological competence but were impaired on some phonetic tasks (presumably indicating the role of “performance” processes). Notably, consistent results were reported by Ramus and Szenkovits [52], who found that phonological deficits in children with dyslexia varied from task to task: they interpreted this as a sign of spare phonological competence. Again, the analysis by Ramus and Szenkovits [52] suggests the role of performance factors, but these authors did not explicitly frame their results within Chomsky’s dichotomy [12].

Overall, the network analysis enabled detecting a domain component for all three abilities tested, i.e., reading, spelling, and maths. Within the multi-level model of learning skills [11], these clusters may indicate the presence of separate competences for these skills. However, caution in drawing this conclusion comes from the observation that task characteristics also tend to cluster within different domains and might contribute to the observed outcome. In future research, it could be interesting to examine the stability of the observed network by introducing as many task variations as possible.

Results also indicated that the three domains were intercorrelated and that some tests covered nodes with a higher strength centrality. Two of these tests (arithmetic facts subtest and spelling words with ambiguous transcription) are particularly interesting in the perspective proposed by the multi-level model of learning skills [11], which is presented in Figure 1. In the model, it is proposed that the ability to consolidate instances acts as the cross-domain factor. Accordingly, it is expected that tasks that heavily call upon the ability to recall specific instances should be correlated independent of their domain. The retrieval of arithmetic facts is a clear example of a task that can only be solved by reference to very specific instances. Much the same is the case of spelling words with ambiguous transcriptions such as “quota” (rate) or “cuoco” (chef), which share a similar sound but differ in their orthographic transcription. To correctly spell these words, the child cannot rely on phoneme-to grapheme conversion rules, but has to recall specific traces, or instances [20,53,54]. It should be noted that the association between these two tasks cannot be easily explained in terms of either competence or performance factors. The two tasks have quite different surface features and also require separate underlying cognitive factors. Therefore, it appears that the strength centrality of these two tasks is coherent with the cross-domain interpretation of the ability to consolidate instances factors, as envisaged by the model.

A third reading task also showed elevated strength centrality, i.e., the decoding accuracy in reading a text passage. It may be observed that the high regularity of the Italian grapheme-to-phoneme conversion rules is such that at least on standard clinical tests, it is somewhat difficult to have conditions which directly call for the activation of lexical entries (i.e., for an instance-based analysis). Thus, for example, irregular words are entirely absent in this language. This does not detract from the important role of lexical activation in this orthography [55], but more simply indicates that isolated measures of lexical activation are comparatively infrequent (although not absent) in standard clinical batteries. Given this premise, it is difficult to advance a definite interpretation of the observation that accuracy in reading a text was also a node with high strength centrality. Along the lines developed above, one might speculate that accuracy is particularly sensitive to lexical information, i.e., that a well-developed lexicon is important to reduce errors in reading. If this is the case, the general interpretation of this marker would be similar to the ones advanced for maths and spelling tasks. Note that reading speed was also associated with both arithmetical fact retrieval and ambiguous word spelling. Indeed, good reading speed might index lexical processing as opposed to serial (and possibly slower) sublexical processing. In this vein, the association among reading, spelling and maths might depend on the ability to consolidate instances. However, note that the network weights between reading speed with arithmetical facts and spelling of ambiguous words were less strong than those between reading accuracy and these variables. One might also consider the possibility that surface similarity among tasks plays a role here; namely, both tests with high strength centrality for maths and spelling (arithmetic facts subtest and spelling words with ambiguous transcription) rely on accuracy (not speed) measures. Thus, it is possible that it was this characteristic that allowed the emergence of the cross-domain role of text reading accuracy, reducing the strength of the network weights with reading speed after the role of reading accuracy was partialled out in the network analysis. Clearly, further research is necessary to clarify this aspect. Indeed, although lexically sensitive conditions are infrequent in reading due to the high regularity of Italian orthography, they are not entirely absent. Therefore, in further confirmations of the network found here, it would be interesting to add conditions, such as orthographic decision tasks, which more specifically require the activation of lexical (i.e., instance-based) traces. 

A further comment regards the application of network analysis in the domain of academic learning skills, (because of its novelty). One advantage of network analysis is that it is exploratory and data-driven. Network analysis is not generally superior to other methods (such as confirmatory factor analysis); however, it fits well with the questions raised in the current study, and the most recent literature on the topic. Research has rarely investigated maths, reading, and spelling skills together; thus, an exploratory approach should make a substantial contribution. Exploratory network analysis does not impose constraints on the model a priori. The network emerges “spontaneously” from the data structure. No latent variables are postulated, and the regularisation parameter is estimated by adopting an automated selector operator. Thus, any model has the potential to emerge. The fact that the one that best fits with the data is coherent with the theoretical model provides strong support for the theoretical model itself. Future studies, adopting a confirmatory approach, could provide additional supportive evidence. In fact, the network and the confirmatory models should not be considered mutually exclusive, but rather should be used to obtain converging evidence from different methodological perspectives and statistical approaches.

Finally, an additional advantage of network analysis is its hypothesis-generating capacity. Although we focused only on specific nodes and relationships, the model estimates all node connections (see Figure 2). Other researchers could identify new hypotheses from these connections which were not the aim of our study, stimulating future research.

## 5. Conclusions

Overall, the network analysis allowed obtaining a data-driven “snapshot” of the learning skills of children with three-to-five years of experience in reading, spelling, and performing calculations. As expected, the obtained network pointed to both elements of association and of partial independence among these academic skills. Interestingly, the present study was based on standard clinical instruments; therefore, it appears that the multi-level model of learning skills [11] might provide a useful framework for the clinical analysis of these academic learning skills. Nevertheless, one should be aware that a definite separation between the hypothesised competence, performance, and acquisition factors requires further work with a larger variety of tasks and on both typically developing and learning-disabled populations.

## Figures and Tables

**Figure 1 brainsci-11-00656-f001:**
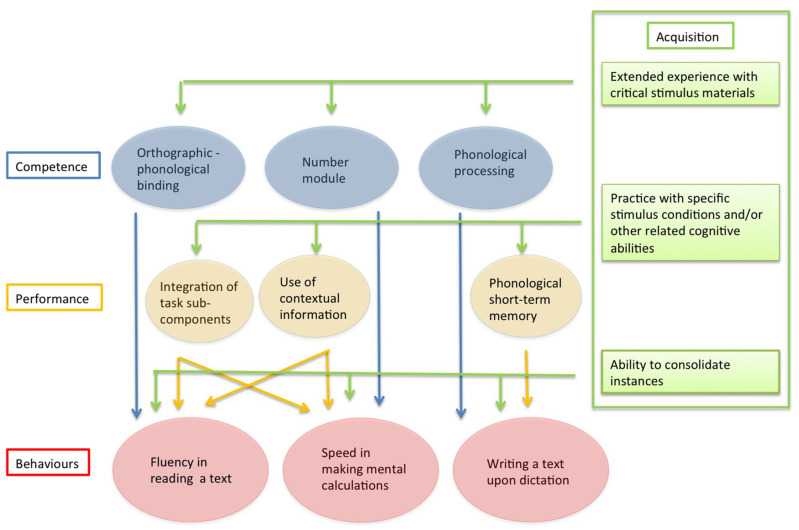
A multi-level model of learning skills [11].

**Figure 2 brainsci-11-00656-f002:**
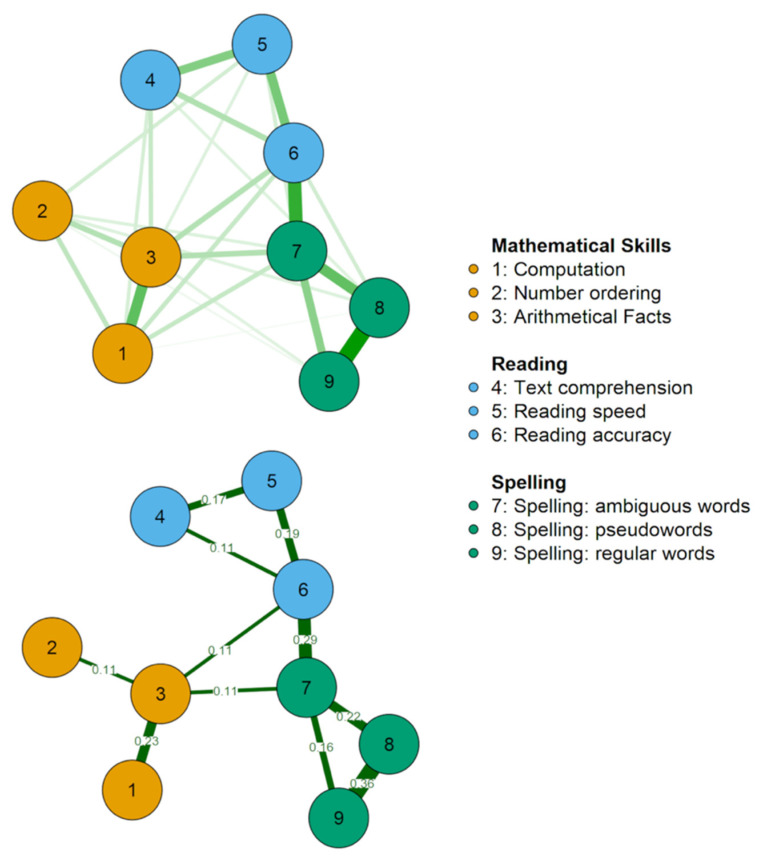
The upper panel reports every edge estimated as different from 0 by the Gaussian Graphical Model, adopting the eBIC Graphical Lasso method. The edges represent regularised partial correlations. The lower panel shows the same network with graphical changes to improve the readability of the most important connections. Specifically, only edges >0.1 are shown. Notably, these edges are also those that emerged as more stable considering the estimated confidence intervals, estimated over 10,000 bootstrap resampling events (see Figure 3). To make it more readable, colour saturation is not scaled according to the edge strength in the lower panel as done in the upper panel. Green lines indicate positive associations. Red lines would have indicated negative associations (none were observed). The nodes indicate the variables as follows: Computation (1); Number ordering (2); Arithmetical Facts (3); Text comprehension (4); Reading speed (5); Reading accuracy (6); Spelling: ambiguous words (7); Spelling: pseudowords (8); Spelling: regular words (9).

**Figure 3 brainsci-11-00656-f003:**
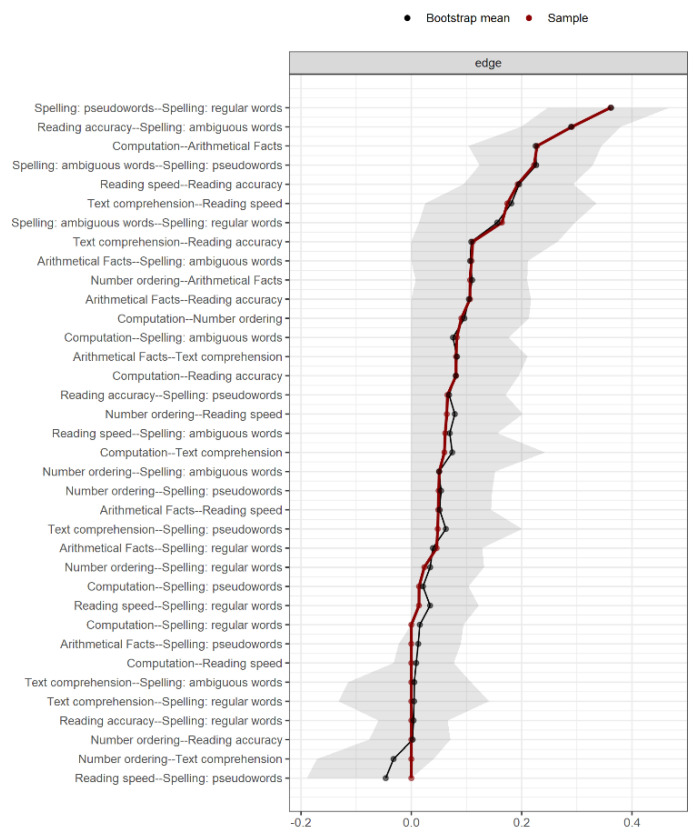
Full list of edges and results from the 1000 bootstraps. Red dots indicate the edge value of the estimated network. Black dots indicate the average edge value over 1000 bootstrap resampling. The grey shadow represents the 95% confidence interval estimated with the bootstrap resampling. Edges are ordered by their estimated strength.

**Table 1 brainsci-11-00656-t001:** Main functions and characteristics of competence, acquisition and performance levels as related to individual differences in learning skills and comorbidity of learning disorders.

	Function(s)	Characteristics	Specificity/Overlap
Competence	Ability to activate a specific set of representations and processes	- Domain-dependent - Task-independent - Sensitive to practice	Dissociation of deficit
Acquisition	- Learning specific rules and/or regularities	- Domain-dependent	
	- Learning direct memory traces (instances)	- Item specific - Domain-independent	Learning disorders across different domains (comorbidity)
	- Learning typical task formats, characteristic of a given behaviour	- Partially domain-dependent	
Performance	Actual performance depends on the characteristics of the task	- Task-dependent - Partially domain-dependent - Sensitive to practice	Both associations and dissociations depending on task similarity

**Table 2 brainsci-11-00656-t002:** The lower part (dark background) reports the network weights, which correspond to regularised partial correlations. The upper part (light background) reports simple correlations, measured with Pearson’s r. The diagonal reports the strength centrality index. The first column reports the mean z scores ± the standard deviations.

Descriptive	Variable	(1)	(2)	(3)	(4)	(5)	(6)	(7)	(8)	(9)
−0.079 ± 0.90	(1) Computation	0.56	0.22	0.37	0.20	0.15	0.27	0.28	0.19	0.16
0.131 ± 0.84	(2) Number Ordering	0.09	0.38	0.24	0.03	0.18	0.14	0.22	0.19	0.17
0.146 ± 1.02	(3) Arithmetical Facts	0.23	0.11	0.72	0.23	0.23	0.32	0.33	0.20	0.21
−0.134 ± 0.90	(4) Text Comprehension	0.06	0.00	0.08	0.47	0.30	0.28	0.18	0.18	0.11
−0.343 ± 0.90	(5) Reading Speed	0.00	0.06	0.05	0.17	0.55	0.36	0.27	0.07	0.16
−0.249 ± 0.96	(6) Reading Accuracy	0.08	0.00	0.11	0.11	0.19	0.85	0.48	0.30	0.22
−0.104 ± 1.22	(7) Spelling: Ambiguous Words	0.08	0.05	0.11	0.00	0.06	0.29	0.98	0.45	0.40
0.064 ± 1.14	(8) Spelling: Pseudowords	0.01	0.05	0.00	0.05	0.00	0.07	0.22	0.76	0.51
−0.325 ± 1.89	(9) Spelling: Regular Words	0.00	0.02	0.05	0.00	0.01	0.00	0.16	0.36	0.61

## Data Availability

Data will be made available only upon request for reasons of privacy.

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
