# Peer review of "A Network Analysis of the Relationship among Reading, Spelling and Maths Skills"

_brainsci, 2021, doi:10.3390/brainsci11050656_

Round 1

Reviewer 1 Report

This is an interesting study, which uses a network modelling method to examine the relationship between different types of learning, relating to language, spelling and mathematics. The approach is clearly described and the findings are presented well. My only real concern relates to the advantages of this approach over other approaches. Also, in comparing this approach to the process models mentioned in the introduction, I wondered whether it would be unfair to describe the present approach as descriptive rather than functional, and to question what advantages this approach has?  It might be useful to briefly outline this in the discussion. 

Author Response

Reviewer 1

This is an interesting study, which uses a network modelling method to examine the relationship between different types of learning, relating to language, spelling and mathematics. The approach is clearly described and the findings are presented well. My only real concern relates to the advantages of this approach over other approaches. Also, in comparing this approach to the process models mentioned in the introduction, I wondered whether it would be unfair to describe the present approach as descriptive rather than functional, and to question what advantages this approach has?  It might be useful to briefly outline this in the discussion.

Response:

Thank you for the appreciation of our study.

We do not believe that the Network analysis is superior to other alternative methods; however, it fits well with the questions of the current study, and the state-of-the-art of the literature on the topic. While a large body of literature investigated these learning skills separately, they have been put together only rarely. A key advantage is that GGM models are exploratory models, which are strongly data-driven. This means that very few constraints have been imposed on the model, which emerge “spontaneously” from the data structure. No latent variables are postulated, and the regularization parameter is estimated by adopting an automated selector operator.

The fact that the emerged model is coherent with the theoretical one represents strong support toward the theoretical model. Certainly, future studies with a confirmatory approach may bring further supportive evidence. Indeed, the network and the confirmatory models should not be considered mutually exclusive, but they may get converging evidence from a different methodological perspective and statistical approach to the same topic.

An additional plus of network analyses is its hypothesis-generating nature. Indeed, while we focused on specific nodes and relations, the model estimates every nodes connection. Other researchers may identify new hypothesis in these “out-of-our-focus” connections.

We added a part of text to specify these aspects in the discussion, as suggested by the reviewer.  We hope that the new presentation proves clearer on these points.

Reviewer 2 Report

Reading, spelling, and arithmetic are central domains of school achievement. There is comprehensive research to each of these domains of scholastic learning, but they are rarely investigated together, which makes it difficult to understand as to why there seem to be strong associations between these learning domains in typical as well as atypical development (in terms of learning disorders). The paper presents an interesting data-driven account in which standard tests assessing different components of each learning domain were tested for associations. This network analysis is based on a sample of 185 unselected children in Grades 3 to 5 and is presented as a first step to test a new multi-level model of learning skills which differentiates between levels of competence, acquisition, and performance.

The topic is thus timely and of great interest. The multi-level model (Table 1 and Figure 1) has been presented in an earlier publication (Zoccolotti et al., 2021), which mostly focused on cognitive predictors of reading, spelling, and arithmetic. In my view it provides an interesting background for the investigation of associations between academic skills and will need further specification over time. I am not familiar with network analysis, but found the approach to investigate the associations between standardized tests assessing different components of reading, spelling, and maths quite revealing. In particular. I liked the conclusion that this analysis provides a (highly interesting) “snapshot” of these associations, based on specific tests and a specific age group.

The finding that the tests cluster according to the scholastic skill they are supposed to measure (reading, spelling, arithmetic) is reassuring. The strong association between spelling (of irregular words) and arithmetic fact retrieval is plausible as explained in the paper – both require recall of specific instances. The link of reading accuracy with these two components is harder to explain, especially in this Italian sample. It is my understanding that Italian has a highly transparent orthographic system so that I would have expected overall high reading accuracy in the assessed age group. Descriptive statistics is not presented. Is the network analysis based on raw scores or standardized scores? (This would be important to report for all tests). If the latter, than it might be the case that a relatively low variance in raw error scores (due to floor effects in the full sample) might get an overly large relevance. It might also be informative to look at the type of errors. Perhaps in the applied text reading paradigm reading errors are mostly of a morphosyntactic kind (e.g. misreading of inflectional word endings)? Such errors might be less decoding-related than language-related. It is important to note that reading accuracy and speed scores are derived from the very same text reading task. It was not fully clear to me from the task description to what extent students were instructed to read the text as quickly as possible. Is it possible that reading errors resulted from time pressure and are thus an indirect measure of reading speed?

Additional points:

  • Consistent order of constructs and tasks across methods and results would increase readability. In the current version, the Maths tasks are presented in the middle in Figure 1, last in the Methods section, first in Figure 2 and Table 2, but last in Figure 2.
  • Participants: Please specify, how exactly intelligence ”within the norm” was defined (what was the minimum score?) and report descriptives.
  • Table 2 is hard to read because of page break. Is reporting 3 post-comma positions for the bivariate correlation of reading speed and reading accuracy a typo?
  • Terminology: In my understanding, the term “learning skills” relates to skills for learning, while in the paper it is used for skills that are learned in school (reading, spelling, math). Would that rather be “learned skills”? But that does not sound appropriate either. I am nor sure I can come up with clearer terminology, but wanted to point this out.
  • The manuscript contains numerous typos and language errors and should be carefully edited.

Author Response

Reviewer 2

Reading, spelling, and arithmetic are central domains of school achievement. There is comprehensive research to each of these domains of scholastic learning, but they are rarely investigated together, which makes it difficult to understand as to why there seem to be strong associations between these learning domains in typical as well as atypical development (in terms of learning disorders). The paper presents an interesting data-driven account in which standard tests assessing different components of each learning domain were tested for associations. This network analysis is based on a sample of 185 unselected children in Grades 3 to 5 and is presented as a first step to test a new multi-level model of learning skills which differentiates between levels of competence, acquisition, and performance.

The topic is thus timely and of great interest. The multi-level model (Table 1 and Figure 1) has been presented in an earlier publication (Zoccolotti et al., 2021), which mostly focused on cognitive predictors of reading, spelling, and arithmetic. In my view it provides an interesting background for the investigation of associations between academic skills and will need further specification over time. I am not familiar with network analysis, but found the approach to investigate the associations between standardized tests assessing different components of reading, spelling, and maths quite revealing. In particular. I liked the conclusion that this analysis provides a (highly interesting) “snapshot” of these associations, based on specific tests and a specific age group.

Response: We thank the reviewer for the appreciation of our work.

The finding that the tests cluster according to the scholastic skill they are supposed to measure (reading, spelling, arithmetic) is reassuring. The strong association between spelling (of irregular words) and arithmetic fact retrieval is plausible as explained in the paper – both require recall of specific instances. The link of reading accuracy with these two components is harder to explain, especially in this Italian sample. It is my understanding that Italian has a highly transparent orthographic system so that I would have expected overall high reading accuracy in the assessed age group. Descriptive statistics is not presented. Is the network analysis based on raw scores or standardized scores? (This would be important to report for all tests). If the latter, than it might be the case that a relatively low variance in raw error scores (due to floor effects in the full sample) might get an overly large relevance. It might also be informative to look at the type of errors. Perhaps in the applied text reading paradigm reading errors are mostly of a morphosyntactic kind (e.g. misreading of inflectional word endings)? Such errors might be less decoding-related than language-related. It is important to note that reading accuracy and speed scores are derived from the very same text reading task. It was not fully clear to me from the task description to what extent students were instructed to read the text as quickly as possible. Is it possible that reading errors resulted from time pressure and are thus an indirect measure of reading speed?

Response: The network was estimated on the scores normalized on the normative data of each tool. Descriptive statistics are now provided as additional columns of table 2.

The distributions of reading speed and accuracy of the sample are appropriate for parametric analysis. We do not have clear evidence of a potential effect of the restriction of range or a ceiling effect. We report here the distribution plots.

Additional information on instructions is presented. In particular:

“In the instructions, children were asked to read as best as possible, i.e., making as few errors as possible and fluently. Children are told that “you have to read as you usually do when you are engaged”.  So, no specific emphasis on reading speeding speed was made.

The accuracy score takes into account the functional meaning of errors. Each word with a substitution, elision, insertion or inversion of letters is scored as 1, while spontaneous self-corrections, errors that do not change the meaning of the text, changes in stress assignment, repetitions of the same errors and hesitations are given a score of ½.  This information has been added to the text.

Please note that this type of scoring, derived from the manual, is focused on evaluating the functional value of reading and does not directly allow responding to questions concerning which type of errors was most frequent in terms of classical psycholinguistic categories.

Finally, while a trade-off of speed over accuracy is plausible, it is unlikely to affect our results. Indeed, we found a positive relation, which means that people reading faster were also more accurate, suggesting a general better proficiency in reading skill.

Additional points:

Consistent order of constructs and tasks across methods and results would increase readability. In the current version, the Maths tasks are presented in the middle in Figure 1, last in the Methods section, first in Figure 2 and Table 2, but last in Figure 2.

Response:

Sorry for this.  We have tried to re-arrange the presentation as suggested by the reviewer.

Figure 1 is reproduction from previous work; so, it cannot be modified.  However, we now present the Maths tasks first in the method section as well as in Figure 2 and Table 2. 

We are not sure about the last expression of the reviewer as it repeats Figure 2 twice.  However, if it referred to Figure 3, please note that, in this case, the order of presentation is due to the output of the bootstrap analysis.

Participants: Please specify, how exactly intelligence ”within the norm” was defined (what was the minimum score?) and report descriptives.

Response:

Children with a performance 2 SD below the mean of the normative sample were excluded from the sample. A total of thirteen children were excluded because of this criterion.  This information has been added to the text, including the descriptive values relative to the performance on the Raven’s Coloured Progressive Matrices.

Table 2 is hard to read because of page break. Is reporting 3 post-comma positions for the bivariate correlation of reading speed and reading accuracy a typo?

Response: Apologize for the typo, we now report only 2 as for any other variable.

Terminology: In my understanding, the term “learning skills” relates to skills for learning, while in the paper it is used for skills that are learned in school (reading, spelling, math). Would that rather be “learned skills”? But that does not sound appropriate either. I am not sure I can come up with clearer terminology, but wanted to point this out.

Response: we have added the adjective academic to the expression learning skills with the idea to better qualify the concept.  We hope that this way of expressing the concept represents an improvement in terms of clarity.

The manuscript contains numerous typos and language errors and should be carefully edited.

Response: The paper has been revised by an English native speaker.  We hope that the new version proves more readable.
